# Sociodemographic Associations of Physical Activity in People of Working Age

**DOI:** 10.3390/ijerph16122134

**Published:** 2019-06-17

**Authors:** Daniel Puciato

**Affiliations:** Faculty of Physical Education and Physiotherapy, Opole University of Technology, ul. Prószkowska 76, 45-758 Opole, Poland; d.puciato@po.edu.pl or d.puciato@po.opole.pl

**Keywords:** physical activity, demographic factors, social factors, working age, city, population studies

## Abstract

The aim of this study was to identify relationships between the physical activity and sociodemographic status of respondents aged 18–64 years. The research was conducted in 2014 and 2015 in Wrocław, Poland. The study group comprised 4460 people. The sample selection was random and stratified. The research tool was the International Physical Activity Questionnaire—Short Form. Levels of physical activity declared by respondents were compared with the recommendations of the American College of Sports Medicine (ACSM). Data on respondents’ sociodemographic status was also obtained. The Mann–Whitney *U* test for samples, Kruskal–Wallis test by ranks, and total and binary logistic regression were used in statistical analysis. Among the respondents, the ACSM health recommendations were met by 43.7% in total (43.2% women and 44.3% men). All analyzed sociodemographic variables differentiated respondents’ physical activity. The youngest respondents were found to be the most physically active. Wrocław residents with a secondary education declared the highest level of physical activity. Among the respondents, manual workers revealed the highest, and the unemployed the lowest odds of meeting the ACSM standards of health-related physical activity. The level of physical activity of unmarried respondents was higher than that of married respondents. The highest percentage of respondents (50.9% women and 54.2% men) with sufficient physical activity levels was found among people living alone. Measures aimed at reducing hypokinesia should be addressed primarily in vulnerable groups, i.e., the unemployed and oldest men.

## 1. Introduction

Reduced physical activity is associated with a growing incidence of lifestyle diseases and premature deaths [1,2]. The effects of hypokinesia are particularly acute for people of working age, as their ability to work depends, to a large extent, on their health status and fitness and performance endurance, which are determined and sustained by physical activity [3]. Therefore, physical exercise of the appropriate volume, frequency, and intensity plays a significant role in both disease prevention and rehabilitation. On the one hand, physical activity contributes to the maintenance or increase of performance capacity and general psychophysical fitness and, on the other hand, it enables the full recovery after diseases, injuries, and fatigue [4].

A low level of physical activity is indirectly related to the occurrence of a number of negative consequences for working-aged people and their families, workplaces, and the entire national economy. For employees, these consequences include, for example, longer working hours and lower work quality, more and longer absenteeism, lower income from work and pensions, and lower quality of life. Reduction of work efficiency, greater turnover of employees and related financial costs, problems with work organization, or declining economic efficiency are the main causes of employees’ hypokinesia for the enterprises employing them. Potential negative effects on the national economy include a shorter life expectancy of physically inactive people, increased healthcare costs, increased spending on sickness benefits, rehabilitation and disability benefits, a lower number of economically active people, reduced state tax revenues and, ultimately, a low economic growth rate [5].

The above problems make it necessary to attach more and more importance to preventive measures aimed at improving people’s lifestyles, including increasing the level of physical activity. However, the development of effective public health plans and strategies must be preceded by diagnostic activities aimed at identifying the main problem areas. With regard to physical activity, this is not only about its overall assessment, but also about its determinants. The results of previous studies show that physical activity is significantly differentiated among various social groups [6,7,8]. Such a diagnosis will make it possible to reach out to groups at particular risk of hypokinesia, and to plan activities aimed at increasing their level of physical activity.

An important group of determinants of physical activity are sociodemographic factors. Despite numerous studies on this issue, the results have been ambiguous and still indicate different directions and strengths of perceived associations [4,9,10,11,12,13]. So far, the number of people in the household has not been considered as a factor influencing the physical activity of working-aged individuals, especially during their professional work, at home, and in leisure time. Few studies have also been conducted on representative samples of local populations. Authors have also rarely examined associations between physical activity and socioeconomic status separately for men and women, while the impact of particular factors on physical activity can differ in respondents of both sexes. 

Considering the above introductory remarks, the aim of this study is to identify relationships between the physical activity and sociodemographic status of respondents aged 18–64 years.

The study addresses the following research problems:What percentage of respondents meets the criteria of health-related physical activity formulated by the American College of Sports Medicine?Do such sociodemographic features as age, level of education, occupation, marital status, and the number of people in the household determine respondents’ compliance with ACSM recommendations for health-related physical activity?Are the strength and directions of relationships between physical activity and sociodemographic variables similar in both the studied women and men?

## 2. Methods

The study was conducted in March and November of both 2014 and 2015 in Wrocław, Poland. The research project had been given positive approval by the Commission of Bioethics of the University School of Physical Education in Wroclaw. The participants were 4460 people, including 2331 women and 2129 men aged 18–64 years. Their characteristics and selected sociodemographic variables are presented in Table 1. The sampling was random (three-stage group variant) and stratified (proportional variant). First, ten out of the alphabetically ranked housing estates of Wrocław were drawn using a random number table, then, following a similar mechanism, three streets were drawn in each housing estate, the inhabitants of which took part in the study. The number of respondents from each housing estate was proportional to the number of its inhabitants. Moreover, the planned sample was divided into layers in which the stratifying features included the respondents’ sex and age. A proportional variant was adopted so that the number of respondents in each subpopulation was proportional to the size of the layers of the population of working aged people from Wrocław.

The minimum sample size was calculated following the formula [14]:
$$n=\frac{N}{1+\frac{4d^{2}\left(N-1\right)}{z_{\alpha }^{2}}}$$
where *n*—minimum sample size; *N*—number of working-aged residents from Wrocław; *d*—standard error of estimate; *z_α_*—critical value of normal distribution at assumed confidence interval.

Restrictive assumptions were made as to the acceptable error of estimate (*d* = 0.02) and confidence interval (α = 0.01). Under these assumptions, the minimum sample size was estimated at 4122 people. During the survey about 10% more questionnaires were collected. After eliminating incomplete or incorrectly completed questionnaires, 4460 people remained in the sample. The number of respondents were taken into account and their random selection for the survey made the sample representative of the population of all Wrocław working-aged residents.

The main research method was a diagnostic questionnaire survey. The International Physical Activity Questionnaire—Short Form (IPAQ-SF) was used [15]. Consideration was given to total physical activity, including physical efforts at work, at home, while commuting, and during leisure time. Respondents’ self-assessments of the volume (duration) and frequency of physical activity at two levels of intensity—high and moderate—were analyzed. The total volume of physical activity expressed in METmin/week was also determined. For this purpose, the MET (metabolic equivalent of task) value corresponding to a given level of physical activity intensity was multiplied by the number of days of physical activity per week and its duration in minutes per day. For example, the intensity of physical activity of an individual practicing jogging four times a week corresponding to 7 MET amounted to 4 × 60 × 7 = 1680 (METmin/week).

The collected data permitted a comparison of the declared physical activity with the recommendations of the American College of Sports Medicine [16]. The ACSM recommendations were chosen because of the lack of Polish norms of physical activity for people in a wide age range and the authors’ intention to compare the obtained results with the results of their previous research. [4,12]. According to the ACSM, it is recommended to undertake physical activity (1) with a high intensity—at least 3 times a week for 20 min a day, or (2) with a moderate intensity—at least 5 times a week for 30 min a day. In the opinion of the authors of the recommendations, this is the minimum level of physical activity, and the implementation of smaller physical efforts may contribute to deterioration of physical fitness and increase the risk of obesity, chronic diseases, and disabilities. Respondents meeting at least one of these criteria were considered to be following the ACSM recommendations. The questionnaire survey also provided information on selected sociodemographic factors, i.e., sex (male, female), age (15–24, 25–34, 35–44, 45–54, 55–64 years), education (primary and basic vocational, secondary, higher), occupation (manual workers, white-collar workers, self-employed, students, unemployed, pensioners), marital status, and the number of people in the household (1, 2, 3, 4, 5, and more).

The obtained data were ordered and subject to structured analysis. The main measures were the number and the percent, which were grouped according to different criteria and listed in tables. The study assessed the asymmetry of the distribution of variables using the asymmetry coefficient (S). Description of the data distribution structure was supplemented with the coefficient of aggregation, kurtosis (K). The test of compliance with the Kolmogorov–Smirnov normal distribution was also carried out. The analysis of test results indicated that the distribution significantly deviated from the standard normal distribution. As a consequence, medians and quartile deviations were calculated. The significance of differences between the total physical activity of male and female respondents in groups separated by age, education, occupation, and the number of people in the household was verified with the Mann–Whitney *U* test for samples of *n* and *m* > 20 while, for marital status, the Kruskal–Wallis test by ranks was used. The significance of differences between particular groups was determined with a post hoc test. The analysis of the relationships between the self-assessment of physical activity by the examined women and men and their sociodemographic characteristics was performed using binary logistic regression. All calculations were performed with the use of the SPSS Statistics 20 software (IBM, Armonk, NY, USA) package. The exante level of statistical significance was set at α = 0.05.

## 3. Results

The total weekly physical activity of the studied women and men was significantly correlated with all sociodemographic variables (*p* < 0.001). The highest mean values of correlation were founded in respondents of both sexes aged 18–34 years, and in men aged 45–54 years. The median energy expenditure was the lowest in women aged 45–54 and in the oldest men. The fact that the average ranks of weekly physical activity in groups separated by age were significantly different was confirmed not only by the value of the Kruskal–Wallis test, but also by the *p*-values for multiple (post hoc) comparisons. The highest level of total physical activity was noted in respondents with a secondary education (2586.0 METmin/week in women, 2994 METmin/week in men), and the lowest in respondents with a primary education (1950.0 METmin/week and 2190.0 METmin/week). The comparison between the post hoc groups showed that statistically significant differences in average values of position measures were found between physical activity of women with a basic vocational education and either a secondary education or higher education, and between men with a basic vocational education and either a secondary and higher education. Considering the occupational status, the most physically active groups of women were students, manual workers, and self-employed, while the most physically active men included pensioners, students, and manual workers. The lowest level of physical activity in women (1752.0 METmin/week) and in men (2072.0 METmin/week) was found among the unemployed. The level of physical activity of women (*Z* = 5.39, *p* < 0.001) and men (*Z* = 5.41, *p* < 0.001) was also determined by their marital status. In both groups of respondents, unmarried individuals were more physically active (2550.0 METmin/week and 2910.0 METmin/week) than married people (2175.0 METmin/week and 2316.0 METmin/week). The median values of energy expenditure of weekly physical activity were the highest in women from one-member (2696.0 METmin/week), two-member (2670.0 METmin/week), and five-and-more-member households (2565.0 METmin/week), and in men from five-and-more (2925.0 METmin/week) and one-member households (2796.0 METmin/week). The least physically active were women from four-member (2034.0 METmin/week) and men from two-member households (2713.1 METmin/week). Statistically significant intragroup differentiation—post hoc test (*p* = 0.02)—was found in women from Wrocław from four- and five-and-more-member households, and between men from two- and three-member households, and from two- and five-and-more member households (Table 2 and Table 3). 

Among the Wrocław residents aged 18–64 years, the ACSM health-related physical activity standards were met by 43.7% of respondents, 43.2% of whom were women and 44.3% were men. Table 3 and Table 4 show the results of logistic regression, revealing associations between ACSM recommendations and respondents’ sociodemographic characteristics. In the group of women, significant associations between physical activity and age were observed in women aged from 35 to 44 years. The odds of their meeting the ACSM recommendations were 33% lower than in the age group of 18 to 24 years. Considering the level of education, the women with a secondary education were almost 80% more likely to fulfill the health-related physical activity recommendations than women with a primary and vocational education. Statistically significant associations between physical activity and occupation were found among manual and white-collar workers, the self-employed, pensioners, and the unemployed. The highest percentage of sufficiently physically active respondents was observed among manual workers (54.2%), and the lowest among the unemployed (30.1%). The odds of meeting the health-related physical activity recommendations were almost three times higher among manual workers (OR = 0.36; CI: 0.26–0.51) than among the unemployed. Almost half of unmarried women and 38.5% of married women were sufficiently physically active. The odds of unmarried women meeting the ACSM standards were almost 40% higher than married women. The number of people in the household was a factor determining the physical activity of women living in one-, two-, and four-person households. Health-related physical activity recommendations were most frequently followed by respondents who lived alone, and the odds of their fulfilling the recommendations was about one-third higher than by women living in two- or four-person households (Table 4).

The highest percentage of sufficiently physically active men in terms of meeting ACSM recommendations was found among the representatives of the two youngest groups of respondents, 63.4% and 55.0%, respectively; and the lowest among the oldest respondents, i.e., 27.2%. The odds of meeting health-related physical activity standards were almost five times higher in the youngest men than in the oldest ones. In men with a secondary education, the odds of meeting the ACSM recommendations were nearly 50% higher than in respondents with a basic and vocational education (OR = 1.48; CI = 1.21–1.82). Statistically significant associations between physical activity and occupation were noted among manual and white-collar workers, the self-employed, and the unemployed. The odds of meeting ACSM standards in the manual workers was almost four times higher than in the unemployed respondents (OR = 0.28; CI = 0.20–0.42). In comparison with married men, the odds of meeting ACSM recommendations were more than twice as high as in single men. The lower limit of 95% confidence interval for odds ratio was 0.39, while the upper limit was 0.55. Among the unmarried men, the recommended physical activity was performed by 55.8%, while among married men, by 37.0%. Similarly to the group of women, this observation was confirmed by the high level of physical activity of men living alone, out of whom 54.2% met the ACSM standards. The odds of respondents from five-person households meeting these recommendations were almost twofold, and from two-person households almost threefold lower, compared to the reference group, i.e., men from one-person households (Table 5).

## 4. Discussion

During a typical week, the majority of respondents performed physical efforts that were insufficient to achieve positive health effects. In addition, an association between physical activity and sociodemographic factors was observed in the study, and groups of respondents with relatively high and particularly low levels of physical activity were identified. The most physically active were the youngest respondents, people with a secondary education, college students, manual workers, unmarried individuals, and respondents from one-member households. The groups of respondents with a particularly low level of physical activity were the unemployed and men from the oldest age group. As they are particularly threatened by hypokinesia, public health programs, including measures aimed at increasing physical activity, should be primarily addressed to them.

In terms of relationships between physical activity and age, it was noted that the most physically active were the youngest respondents of both sexes and the oldest women, while the least active were respondents aged 35–44 years and the oldest men. While the relatively high physical activity in young people and its decrease with age have been reported in the literature [4,8,12,17,18], the noted high level of physical activity in the oldest women and the low level in the age group of 35–44 year-olds have been rarely found. However, Moniruzzaman et al. [19] also observed relatively high levels of physical activity in urban women aged 55–64 years. This may be due to the fact that the workload and household duties of women at this age are often lower due to their retirement and their children living on their own. Still, good health and quite common participation of women in organized forms of leisure activities, e.g., universities of the third age, allow them to take part in relatively intensive physical activities [20]. It should be emphasized, however, that relatively high levels of physical activity of older people were not found in earlier studies. This may have been caused by the fact that in the group of women aged 55–64, there was an overrepresentation of people with normal body structure and good physical and health condition. Rocha et al. [21] also reported the lowest percentage of leisure time physical activity among 35–46-year-olds. However, Nawrocka et al. [7] found no significant correlations between physical activity and age. An attempt to explain this phenomenon could be that the fact that 30–50 years of age is often connected with the dynamic development of one’s professional career, family life, and social activity. It may be connected with limited free time and decreasing motivation to undertake physical exercises. Moreover, in the current economic reality, it is also often a period of debt repayments resulting from the purchase of an apartment or a house, its furnishings, etc. This further intensifies the pressure to increase working time at the expense of reduced leisure time. This interpretation of low physical activity of people aged 35–44 years is also supported by the fact that only one in four respondents performed physical work. In most cases, an increase in working hours was not associated with an increase in physical activity. Not without significance may also be the health problems prevalent in people at this age, e.g., obesity, spinal pain, or emotional exhaustion, which are not conducive to undertaking physical activity [22,23,24].

Respondents’ education level turned out to be a significant social factor determining physical activity. The respondents of both sexes with a secondary education were characterized by a higher level of total physical activity and fulfilled the ACSM standards more often than those with a primary and basic vocational education. Positive correlations between physical activity and the level of education had been found by Kwaśniewska et al. [8], Choi et al. [25], Msambichaka et al. [26], and Rocca et al. [27]. Our earlier research on the Katowice population [4,12] and studies by Biernat [9] and Gubelmann [28] indicated a decrease in physical activity with an increase in the level of education. The current study, however, revealed a rare phenomenon of the highest physical activity level in respondents with a secondary education. This may demonstrate that in today’s urban environment, the secondary level of education has already been associated with sufficiently high cultural, social, and economic capital and its benefits, also in relation to health behaviors including health-related physical activity. Jurakic et al. [29] also showed that the amount of undertaken physical exercise was positively correlated with educational attainment. The education structure and labor market requirements may also significantly contribute to the fact that a large number of people with a secondary education perform physical work, which may affect the respondents’ total physical activity analyzed in the present study.

In the Wrocław population, the highest probability of meeting physical activity standards among manual workers—as also previously noted among adult inhabitants of Katowice [4]—was confirmed. Similar results were obtained by Chen et al. [30]. Most likely, the high total physical activity of this occupational group is mainly due to their high physical activity levels unrelated to exercise. Mirecz et al. [31] also demonstrated that physical activity was one of the most important predictors of mental health and employees’ ability to work in executive positions. This may also motivate manual workers to undertake physical efforts outside their occupational field. Low physical activity among non-working people was also documented in the literature [4,19,32], possibly determined by a lack of physical activity at work, limited financial resources, and the negative psychological and social consequences of unemployment. Higher physical activity levels were noted in unmarried than married persons. Similar observations were made by Basset et al. [32], according to whom one of the main reasons for this was mostly high physical activity associated with single people than in people in relationships. Individuals living alone, with no family life, have more free time, some of which they may use for physical exercise. This is confirmed by the relatively high percentage of male and female respondents from one-person households meeting the ACSM recommendations. Slightly surprising, however, is the noted lowest level of physical activity among respondents from four-member households (women) and two-member households (men). These groups were characterized by a lower level of physical activity than individuals living alone, as well as than respondents from three- or even five-or-more-member households. This factor has not been previously analyzed in studies of working-age populations. It was considered only by Peralta et al. [33] and only in relation to the elderly. These authors found the highest physical activity among people over 65 years living in households with five or more persons, and the lowest among people living alone. The high levels of physical activity of respondents from households with the largest number of people may be due to several reasons. The first one is the high intensity of physical activity in their professional work and house duties, which is probably necessary for the proper functioning of a large family. At present, well-paid jobs are also common in industrial labor posts which require significant physical contributions from employees. The second is the observation that large households are not always households with many children. Sometimes these are simply multigenerational families in which grandparents can financially support parents or help them carry out household duties or care for children. This offers them quite good prospects in terms of undertaking leisure time physical activity. On the other hand, two-person households are not only childless married couples or domestic partnerships, but also single parents with a child. While for childless couples, the possibilities of spending free time seem to be great, it can be quite different for single parents. For similar reasons, the low physical activity of women from four-person households can be observed.

The present study has its strengths and limitations. The strengths certainly include a research sample which is representative of Wrocław, and an analysis of physical activity among representatives of various sociodemographic groups. The analysis of occupational status as a modifier of physical activity should be considered particularly valuable, as previous studies have rarely compared manual workers, white-collar workers, entrepreneurs, school and college students, retired people, and the unemployed. The analysis of relations between the level of physical activity and the number of people in the household, which had not been carried out before among people in this age bracket, is also innovative in the context of obtained results. The relatively high physical activity in the oldest age group of women and the positive role of secondary education as a modifier of total physical activity are interesting observations that have not been accounted for in earlier research. On the other hand, limitations of the study included use of the short form of the physical activity questionnaire, and lack of data on respondents’ body weight, physical fitness, and health status, the cross-sectional nature of research, and the limitation of the spatial scope to one city. The main drawbacks of the short version of the IPAQ questionnaire, as compared to objective methods of physical activity measurement, include difficulty in estimating the intensity and duration of physical activity, problems with remembering on which days of the week respondents engaged in physical activity, and a tendency to overestimate the level of their physical activity [34,35]. Future studies should use tools enabling not only a comprehensive but also a separate analysis of physical activity in different areas of human life, i.e., at work, at home, in leisure time, or while commuting. They should also include the measurement of somatic features and the assessment of respondents’ physical fitness and health status. This will allow for an in-depth analysis of test results and an explanation of possible artefacts, e.g., the lack of a linear relationship between physical activity and the age of respondents. The continuity of such research and extending its spatial scope not only to other large cities but also to smaller towns and villages are definitely worth considering. This approach will allow researchers to create a specific map of spatial units and sociodemographic groups, especially those threatened by hypokinesia. Thanks to systematic research, it will also be possible to identify trends in physical activity which seem to be particularly important due to the dynamic and turbulent changes currently taking place in Central Europe and worldwide.

## 5. Conclusions

The level of physical activity of the majority of respondents was insufficient to obtain significant health benefits.Age, level of education, occupation, marital status, and the number of people in the household significantly differentiated respondents’ physical activity.The highest physical activity was observed in persons aged up to 34 years, with a secondary education, manual workers, college students, single respondents, and respondents living in one- member households.The unemployed respondents of both sexes and the oldest men are two groups at particular risk of hypokinesia. Consequently, programs aimed at physical activity improvement should be addressed primarily to these groups.In the studied group, there were not many differences in the directions and strength of physical activity and sociodemographic characteristics in respondents of both sexes. The exceptions were the high physical activity of respondents aged 55–64 years, found only in the group of women, and the lowest level of physical activity among women from four-person households and men from two-person households.

## Figures and Tables

**Table 1 ijerph-16-02134-t001:** Number and structure of respondents grouped according to sociodemographic characteristics.

Variable	Total (*n* = 4460)		Women (*n* = 2331)		Men (*n* = 2129)	
	*n*	%	*n*	%	*n*	%
**Age (years)**						
15–24	620	13.9	317	13.6	303	14.2
25–34	1189	26.7	609	26.1	580	27.2
35–44	882	19.8	444	19.0	438	20.6
45–54	756	17.0	396	17.0	360	16.9
55–64	1013	22.6	565	24.3	448	21.1
**Education**						
primary and basic vocational	1740	39.0	802	34.4	938	44.1
secondary	1617	36.3	996	42.7	621	29.2
higher	1103	24.7	533	22.9	570	26.7
**Occupation**						
manual worker	1165	26.1	424	18.2	741	34.8
white-collar worker	1356	30.4	764	32.8	592	27.8
self-employed	616	13.8	216	9.3	400	18.8
student	637	14.3	441	18.9	196	9.2
pensioner	283	6.3	240	10.3	43	2.0
unemployed	403	9.1	246	10.5	157	7.4
**Marital status**						
unmarried	1828	41.0	1000	42.9	828	38.9
married	2632	59.0	1331	57.1	1301	61.1
**Number of people in the household**						
1	561	12.6	218	9.4	343	16.1
2	1003	22.5	564	24.2	439	20.6
3	1264	28.3	675	29.0	589	27.7
4	1173	26.3	622	26.7	551	25.9
≥5	459	10.3	252	10.7	207	9.7

**Table 2 ijerph-16-02134-t002:** Differences in energy expenditure of total weekly physical activity of women with regard to sociodemographic characteristics.

Variable	Category	Energy Expenditure of Physical Activity (METmin/week)				Test Post Hoc—*p*-Value					
		Me	Q	H (Z)	*p*	(1)	(2)	(3)	(4)	(5)	(6)
Age (years)	(1) 18–24	2430.0	1165.0	42.2	0.00	x	1.00	0.04	0.00	0.00	x
	(2) 25–34	2565.0	1404.2			1.00	x	0.00	0.00	0.00	x
	(3) 35–44	1980.0	1147.5			0.04	0.00	x	1.00	1.00	x
	(4) 45–54	1752.0	1042.5			0.00	0.00	1.00	x	1.00	x
	(5) 55–64	1880.0	1181.0			0.00	0.00	1.00	1.00	x	x
Education	(1) primary	1950.0	1200.0	47.1	0.00	x	0.00	0.00	x	x	x
	(2) secondary	2586.0	1392.3			0.00	x	1.00	x	x	x
	(3) higher	2556.0	1802.2			0.00	1.00	x	x	x	x
Occupation	(1) unemployed	1752.0	845.0	65.0	0.00	x	0.00	0.45	0.00	0.00	0.00
	(2) student	2748.0	1357.7			0.00	x	0.00	1.00	0.00	1.00
	(3) pensioner	1950.0	1520.5			0.45	0.00	x	0.00	1.00	0.06
	(4) worker	2785.0	1428.0			0.00	1.00	0.00	x	0.00	1.00
	(5) white collar	2100.0	1438.2			0.00	0.00	1.00	0.00	x	0.19
	(6) self-employed	2641.5	1227.0			0.00	1.00	0.06	1.00	0.19	x
Marital status	(1) unmarried	2550.0	1341.0	5.3	0.00	x	x	x	x	x	x
	(2) married	2175.0	1379.0			x	x	x	x	x	x
Number of people in the household	(1) 1	2696.0	1502.2	13.7	0.00	x	1.00	0.82	1.00	0.18	x
	(2) 2	2670.0	1210.0			1.00	x	1.00	0.38	1.00	x
	(3) 3	2455.0	1387.5			0.82	1.00	x	0.08	1.00	x
	(4) 4	2034.0	1174.0			1.00	0.38	0.08	x	0.02	x
	(5) ≥5	2565.0	1331.2			0.18	1.00	1.00	0.02	x	x

Abbreviations: Me—median, Q—quartile deviations, H—Kruskal–Wallis test, Z—Mann–Whitney test, *p*—test probability value, x—the item cannot be filled in due to the arrangement of the table, or it is not possible or intended to be filled in.

**Table 3 ijerph-16-02134-t003:** Differences in energy expenditure of total weekly physical activity of men with regard to sociodemographic characteristics.

Variable	Category	Energy Expenditure of Physical Activity (METmin/week)				Test Post Hoc—*p*-Value					
		Me	Q	H (Z)	*p*	(1)	(2)	(3)	(4)	(5)	(6)
Age (years)	(1) 18–24	3386.0	1344.0	97.3	0.00	x	0.00	0.00	0.00	0.00	x
	(2) 25–34	3066.0	1653.3			0.00	x	0.00	1.00	0.00	x
	(3) 35–44	2316.0	1246.5			0.00	0.00	x	0.00	1.00	x
	(4) 45–54	3105.0	1713.3			0.00	1.00	0.00	x	0.05	x
	(5) 55–64	2031.0	1518.0			0.00	0.00	1.00	0.05	x	x
Education	(1) primary	2190.0	1680.0	25.2	0.00	x	0.00	0.11	x	xx	x
	(2) secondary	2994.0	1672.5			0.00	x	0.03	x	x	x
	(3) higher	2649.0	1555.7			0.11	0.03	x	x	x	x
Occupation	(1) unemployed	2072.0	720.0	58.0	0.00	x	0.00	0.00	0.00	0.47	1.00
	(2) student	3220.5	1248.3			0.00	x	1.00	0.05	0.00	0.00
	(3) pensioner	3390.0	1302.0			0.00	1.00	x	1.00	0.11	0.02
	(4) worker	2745.0	1776.0			0.00	0.05	1.00	x	0.03	0.00
	(5) white collar	2346.0	1404.3			0.47	0.00	0.11	0.03	x	1.00
	(6) self-employed	2316.0	1773.0			1.00	0.00	0.02	0.00	1.00	x
Marital status	(1) unmarried	2910.0	1528.8	5.41	0.00		x	x	x	x	x
	(2) married	2316.0	1584.0				x	x	x	x	x
Number of people in the household	(1) 1	2796.0	1885.5	16.6	0.00	x	0.05	1.00	1.00	1.00	x
	(2) 2	2072.0	1393.5			0.05	x	0.02	0.32	0.00	x
	(3) 3	2499.0	1588.0			1.00	0.02	x	1.00	1.00	x
	(4) 4	2613.0	1194.0			1.00	0.32	1.00	x	0.44	x
	(5) ≥5	2925.0	1687.5			1.00	0.00	1.00	0.44	x	x

Abbreviations: Me—median, Q—quartile deviations, H—Kruskal–Wallis test, Z—Mann–Whitney test, *p*—test probability value, x—the item cannot be filled in due to the arrangement of the table, or it is not possible or intended to be filled in.

**Table 4 ijerph-16-02134-t004:** Physical activity and selected sociodemographic associations in studied women from Wrocław (*n* = 2331).

Variable	ACSM				OR	CI	
	Yes		No				
	*n*	%	*n*	%		−95%	95%
**Age (years)**							
18–24	139	43.8	178	56.2	Ref.	-	
25–34	291	47.8	318	52.2	1.172	0.89	1.54
35–44	152	34.2	292	65.8	0.667	0.50	0.90
45–54	160	40.4	236	59.6	0.868	0.64	1.17
55–64	266	47.1	299	52.9	1.139	0.86	1.50
**Education**							
primary and basic vocational	293	36.5	509	63.5	Ref.	-	
secondary	503	50.5	493	49.5	1.772	1.47	2.14
higher	212	39.8	321	60.2	1.147	0.92	1.44
**Occupation**							
manual worker	230	54.2	194	45.8	Ref.	-	
white collar worker	292	38.2	472	61.8	0.522	0.41	0.66
self-employed	95	44.0	121	56.0	0.662	0.48	0.92
student	220	49.9	221	50.1	0.840	0.64	1.10
pensioner	97	40.4	143	59.6	0.572	0.42	0.79
unemployed	74	30.1	172	69.9	0.363	0.26	0.51
**Marital status**							
unmarried	496	49.6	504	50.4	Ref.	-	
married	512	38.5	819	61.5	0.635	0.54	0.75
**Number of persons in the household**							
1	111	50.9	107	49.1	Ref.	-	
2	223	39.5	341	60.5	0.630	0.46	0.86
3	309	45.8	366	54.2	0.814	0.60	1.10
4	249	40.0	373	60.0	0.644	0.47	0.88
≥ 5	116	46.0	136	54.0	0.822	0.57	1.18

Abbreviations: OR—odds ratio; CI—confidence interval; Ref.—reference groups; ACSM Yes—respondents meeting ACSM recommendations; ACSM No—respondents not meeting ACSM recommendations.

**Table 5 ijerph-16-02134-t005:** Physical activity and selected sociodemographic associations in studied men from Wrocław (*n* = 2129).

Variable	ACSM				OR	CI	
	Yes		No				
	*n*	%	*n*	%		−95%	95%
**Age (years)**							
18–24	192	63.4	111	36.6	Ref.	-	
25–34	319	55.0	261	45.0	0.707	0.53	0.94
35–44	137	31.3	301	68.7	0.263	0.19	0.36
45–54	173	48.1	187	51.9	0.535	0.39	0.73
55–64	122	27.2	326	72.8	0.216	0.16	0.30
**Education**							
primary and basic vocational	396	42.2	542	57.8	Ref.	-	
secondary	323	52.0	298	48.0	1.484	1.21	1.82
higher	224	39.3	346	60.7	0.886	0.72	1.10
**Occupation**							
manual worker	402	54.3	339	45.7	Ref.	-	
white collar worker	215	36.3	377	63.7	0.481	0.39	0.60
self-employed	137	34.3	263	65.8	0.439	0.34	0.57
student	120	61.2	76	38.8	1.332	0.97	1.84
pensioner	29	67.4	14	32.6	1.747	0.91	3.36
unemployed	40	25.5	117	74.5	0.288	0.20	0.42
**Marital status**							
unmarried	462	55.8	366	44.2	Ref.	-	
married	481	37.0	820	63.0	0.465	0.39	0.55
**Number of persons in the household**							
1	186	54.2	157	45.8	Ref.	-	
2	131	29.8	308	70.2	0.359	0.27	0.48
3	280	47.5	309	52.5	0.765	0.59	1.00
4	263	47.7	288	52.3	0.771	0.59	1.01
≥ 5	83	40.1	124	59.9	0.565	0.40	0.80

Abbreviations: OR—odds ratio; CI—confidence interval; Ref.—reference groups; ACSM Yes—respondents meeting the ACSM recommendations; ACSM No—respondents not meeting the ACSM recommendations.

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
