# Peer review of "Sociodemographic Associations of Physical Activity in People of Working Age"

_ijerph, 2019, doi:10.3390/ijerph16122134_

Round 1
Reviewer 1 Report
General comments
The manuscript presents a study of socio-demographic correlates of physical activity in working-age people in Wroclaw, Poland. The evaluation of physical activity is achieved using the short form of the IPAQ, which has major limitations compared to objective measures of physical activity. This needs to be made clearer in the discussion. There are also improvements required in the statistical analysis. The results should include comparisons between groups within each socio-demographic category, rather than simply stating which group had highest and lowest physical activity levels.
Specific comments
Abstract
The most important results should be presented in the abstract using figures, rather than summary sentences.
Methods
- Although the use of a stratified random sampling technique is a strength, it would be worthwhile presenting more details of how this was achieved.
- Non-parametric statistical tests were used throughout, presumably because the data was not normally distributed. However, no details of tests of normality are reported.
- More details are required about the statistical tests used. For instance, specific of the regression need to be provided. Why were results for men and women reported separately, rather than adjusting for sex in a combined analysis?
Results
- Means and SD were used to report energy expenditure grouped by socio-demographic variables (Table 2). Given the apparent non-normality of the data, it would be better to report medians and inter-quartile ranges.
- Throughout the results section, lowest and highest values of energy expenditure for each group were reported, however the statistical tests used were only for the overall group effect. This means that no individual values were compared, therefore it’s not accurate to say, for instance, that the lowest energy expenditure for men was reported in the 35-44 year age group when this value (2511 ± 2059 MET min/week) is almost certainly not significantly different from that of 55-64 year old men (2636 ± 2184 MET min/week).
- There are some errors in reporting the results in the text, with Line 179 of page 6 stating that men with a secondary education were almost half as likely to meet ACSM recommendations as men with a primary or vocational education, when this should be almost 50% more likely, according to Table 4.
- Likewise, the text on line 188 of page 6 states that 55.8% of married men met ACSM guidelines compared to 37% of single men, whereas in Table 4, the values are reversed (unmarried 55.8%, married 37%).
Discussion
- The use of the short form of the IPAQ, a subjective measure of physical activity, is a major limitation of the study. This is only briefly mentioned in the discussion as a weakness, but with no further explanation provided. Subjective measures of physical activity habitually over-report physical activity in comparison to objective measures such as accelerometers. This means that the findings need to be interpreted with caution.
Author Response
Response to Reviewer 1's review
Dear Reviewer,
I would like to thank you very much for your insightful assessment and all the comments, especially those critical of my manuscript. I have made all the suggested changes and additions, and I believe that they will greatly improve the quality of my manuscript.
As suggested, numbers have been used to describe the main research results. The Methods section describes now in more detail the process of random sample selection, the course and results of the study of variable distributions, and it also justifies a separate analysis of the relationships between physical activity and socio-demographic factors in groups divided by sex. In Results, due to the lack of conformity of data with the normal distribution, arithmetic means and standard deviations have been replaced with medians and quartile deviations. The analysis study results is also supplemented by the results of intra-group comparisons, therefore Table 2 has been divided into Tables 2 and 3, according to sex. In the description of the logistic regression analysis incorrect interpretations of the research results have been corrected (lines 179 and 188, page 6). The Discussion section includes a limitation related to the use of the short version of the IPAQ questionnaire and the main drawbacks of this research tool are described. In this context, some research results have been interpreted more cautiously.
All changes are indicated in the text. Once again, thank you very much for reviewing my work and all the comments. I hope that the revised article will be satisfactory. Otherwise, I am ready to continue working on the text.
Please accept the assurance of my highest consideration,
The Author

Reviewer 2 Report
General comments:
-I really like this paper and advocate for revise and resubmit based on the comments below.
-Does Poland have any recommended guidelines for physical activity? Is ACSM the only guideline examined?
-Results from logistic regression are not just correlations, they are associations that provide magnitude and directionality. I suggest replacing the word “correlation” with “association”.
-Body weight, physical function, and disease status should be added as covariates if the data contain these variables. If the older adult sample is primarily made up of older adults who are lower weight, higher functioning, and without disease, then that will be one explanation to why they have higher levels of physical activity. If these estimates are adjusted for the above covariates and the results remain similar to the original results, then we can be confident these are actually important socio-demographic factors of physical activity of the working class.
Title:
-This is a grammatical question but is it “working-age” or “working-aged”? Other than that, title is fine.
Abstract:
-The first two sentences could probably be removed.
-The age range is repeated and could be removed the second time to save words.
-Why was the sample selection random? Are the authors referring to recruitment or the analytic sample?
-Instead of noting ACSM guidelines, can the authors report what the actual ACSM guidelines entail (150 minute of MVPA per week)?
-There are no actual results reported (e.g. numbers). I would like to see estimates and p-values from the models for each finding stated.
-The discussion is lacking – the entire population is largely sedentary, we cannot just focus on sub groups. However, these physical activity correlates identify factors that may associate with even more physical inactivity and suggest tailored interventions.
Introduction:
-I would be careful noting reduced physical activity as a cause of “lifestyle disease” and premature mortality. The evidence is mixed largely because of the different ways to measure physical activity (e.g., older adults may expend more energy to perform a “light” intensity physical activity task common to middle-aged adults). Also, low physical activity can be a result of disease and an indicator of impending mortality. However, as the authors know, there is a large body of literature showing associations between physical activity and health. This is where I suggest softening the opening sentence (e.g., replace cause with association).
-Need a reference for the first sentence of the second paragraph.
Methods:
-The sample size consideration is a nice addition!
-In table 1, column 1, can the authors add bullets or dashes next to sub-rows? For example, under “Education”, can dashes be added to the levels of education? This will enhance readability.
-Can the authors add a table or supplementary table of the actual questions used from IPAQ-SF?
-Please add ethical considerations such as obtaining written informed consent and institutional review board approval (or equivalent entity).
-Are the authors running all the factors within one model or running separate models?
Results:
-The authors report ACSM guidelines as minutes needed per week in certain intensities. However, the results are not in minutes per week but rather MET*min per week. Either the authors need to change Table 2 into minute per week in certain intensities or provide information for how much MET*min/week is considered reaching ACSM guidelines. It is difficult for a reader to reach energy expenditure and then try to see whether analytic sample averages are near guideline recommendations.
-The results are a bit confusing. If I look at table 2 and just visualize the numbers, the highest group is just men aged 18-24. However, the authors report that women aged 25-34, 55-64 and men 18-34 all have the highest mean total physical activity. There was no statistical test reported between these specific age groups. The only p-values I see are testing whether levels within each factor are different (unless I am missing something).
-For table 3, can the authors label the reference groups for each factor? All I see is OR=1.000 because remember that OR=1.000 can also be a result. I suggest replacing with “reference” to enhance readability.
Discussion:
-I really like that the authors provide a pretty comprehensive discussion on the findings.
-The authors should reiterate sub groups who had the lowest physical activity levels and the overall prevalence of meeting ACSM guidelines in the first paragraph for readers.
-It is odd to me that the authors did not find a linear relationship of self-reported physical activity decline with age but I sincerely believe that this is measurement error of self-report physical activity. Also, with age, measured energy expenditure needed to perform daily activities tends to go up with age (see ref. below). Therefore, the argument that more lower intensity activities are more prevalent in older women needs counterintuitive as doing chores may actually bump up an older adult into performing moderate intensity physical activity.
Knaggs, J. D., Larkin, K. A., & Manini, T. M. (2011). Metabolic cost of daily activities and effect of mobility impairment in older adults. Journal of the American Geriatrics Society, 59(11), 2118-2123.
-The second to last paragraph is a bit long. Can the authors segment into smaller paragraphs?
-I would not call the limitations of the study “weaknesses”. This is a good study and every study has limitations. However, the authors should provide a little more on the limitations of self-reported physical activity and that social and recall biases can heighten measurement error.
-I note above that individual factors such as weight characteristic, functional status, and disease status are not accounted for. These are limitations if these are not added to models. Also, the models are not adjusted for all the factors at once (except for sex). This is another limitation.
Author Response
Response to Reviewer 2's review
Dear Reviewer,
I would like to thank you very much for your insightful assessment and all the comments related to my manuscript, especially the critical ones. I have made all the suggested revisions, and I believe they will greatly improve the quality of my text. The title of the work has been corrected, the first two sentences in the Abstract and the repeated information on the age bracket have been removed, and 'correlation' has been replaced with 'association' throughout the whole text. In the Introduction, the word "cause" has been replaced by the word "associated". As suggested a provision on the consent of the relevant bioethics committee to conduct the research has been included in the Methods section. The descriptions of the random sample selection process, the IPAQ questionnaire, and applied statistical analysis methods have been made in a more detailed way, and an example of calculating the energy expenditure of the total weekly physical activity of the respondents expressed in METmin/week has been provided. The choice of criteria for the assessment of respondents’ physical activity (ACSM) has been also justified. The analysis of research results is supplemented by the results of intra-group comparisons, therefore Table 2 has been divided into Tables 2 and 3, according to respondents’ sex. Tables 4 and 5 clearly define reference groups. The Discussion includes the most important research results, emphasizes the low level of physical activity of the entire population, and indicates the groups with the highest and lowest levels of physical activity. The suggestion that the high physical activity of oldest women was probably related to low and medium intensity efforts has been removed. Also an observation has been added that a cautious approach to the artefacts obtained in the study, i.e. lack of linear dependence of physical activity on age and the high level of physical activity in older women, is necessary. These findings could also be associated with over-representation of people with normal body weight and good physical and health condition among women aged 55-64 years. The lack of such data was considered one of the limitations of the study and a conclusion has been formulated for the necessity to collect such data in subsequent studies. The limitations of the study include now the weaknesses of the applied research tool in the context of measurement precision.
All the changes have been marked in the text. Once again, thank you very much for reviewing my manuscript and all the comments. I hope that the amended manuscript will be satisfactory. Otherwise, I am ready to continue working on the text.
Please accept the assurance of my highest consideration.
The author
